# Receptor-mediated yolk uptake is required for *oskar* mRNA localization and cortical anchorage of germ plasm components in the *Drosophila* oocyte

Tsubasa Tanaka[1,2]*, Naoki Tani[3], Akira Nakamura[1,2]*

1 Department of Germline Development, Institute of Molecular Embryology and Genetics, Kumamoto University, Kumamoto, Japan, 2 Graduate School of Pharmaceutical Sciences, Kumamoto University, Kumamoto, Japan, 3 Liaison Laboratory Research Promotion Center, Institute of Molecular Embryology and Genetics, Kumamoto University, Kumamoto, Japan

* ttsubasa@kumamoto-u.ac.jp (TT); akiran@kumamoto-u.ac.jp (AN)

**Data Availability Statement:** All relevant data are within the paper and its Supporting Information files.

## Abstract

The *Drosophila* germ plasm is responsible for germ cell formation. Its assembly begins with localization of *oskar* mRNA to the posterior pole of the oocyte. The *oskar* translation produces 2 isoforms with distinct functions: short Oskar recruits germ plasm components, whereas long Oskar remodels actin to anchor the components to the cortex. The mechanism by which long Oskar anchors them remains elusive. Here, we report that Yolkless, which facilitates uptake of nutrient yolk proteins into the oocyte, is a key cofactor for long Oskar. Loss of Yolkless or depletion of yolk proteins disrupts the microtubule alignment and *oskar* mRNA localization at the posterior pole of the oocyte, whereas microtubule-dependent localization of *bicoid* mRNA to the anterior and *gurken* mRNA to the anterior-dorsal corner remains intact. Furthermore, these mutant oocytes do not properly respond to long Oskar, causing defects in the actin remodeling and germ plasm anchoring. Thus, the yolk uptake is not merely the process for nutrient incorporation, but also crucial for *oskar* mRNA localization and cortical anchorage of germ plasm components in the oocyte.

## Introduction

Asymmetric localization of specific RNAs and proteins in developing oocytes is fundamental for body patterning and cell fate determination of future embryos in a variety of organisms. In the *Drosophila* oocyte, specific maternal factors that direct abdominal patterning and germ cell formation are localized in the specialized cytoplasm at the posterior pole, called germ (pole) plasm. The germ plasm is assembled by stepwise targeting of its components to the posterior pole of the oocyte. Germ plasm components are stably maintained at the posterior cortex until their inheritance by germline progenitors (pole cells) in the early embryo [1].

Assembly of the germ plasm begins with localization of *oskar* (*osk*) mRNA to the posterior pole of the oocyte [2]. Transport of *osk* mRNA in the oocyte depends on polarized microtubule arrays whose plus-ends are weakly enriched at the posterior pole [3,4]. Once localized, *osk*

**Funding:** This work was supported in part by Grant-in-Aids for Scientific Research from the Japan Society for the promotion of Science (KAKENHI number 26840082 and 16K07374 to TT, and 25650088, 26114508 and 17H03686 to AN), the program of the Joint Usage/Research Center for Developmental Medicine, Inter-University Research Network for Trans-Omics Medicine, Institute of Molecular Embryology and Genetics, Kumamoto University to TT, and research grant from Takeda Science Foundation and Mitsubishi Foundation to AN. The funders had no role in study design, data collection and analysis, decision to publish, or preparation of the manuscript.

**Competing interests:** The authors have declared that no competing interests exist.

**Abbreviations:** ApoB, Apolipoprotein B; Apolpp, Apolipophorin; Aub, Aubergine; Capu, Cappuccino; Kin-βgal, Kinesin-β-galactosidase; LDLR, low-density lipoprotein receptor; osk, oskar; Spir, Spire; Stau, Staufen; Tud, Tudor; Vas, Vasa; Yl, Yolkless.

mRNA produces 2 isoforms, long and short Osk, through alternative usages of 2 in-frame start codons. Long Osk shares the entire short Osk moiety, but they have distinct functions [5–8]. Short Osk nucleates germ plasm assembly by recruiting downstream components of the germ plasm, such as Vasa (Vas), Tudor (Tud), and Aubergine (Aub), as well as germ plasm mRNAs that encode factors involved in the establishment of the germ cell fate [9]. In contrast, long Osk has almost no activity to recruit germ plasm components but is indispensable for anchoring them to the oocyte cortex. In the oocytes lacking long Osk, components of the germ plasm including short Osk diffuse back into the ooplasm. The failure in the germ plasm anchoring to the cortex causes severe reduction in the number of pole cells formed at the posterior pole [8]. The 2 Osk isoforms also differ in their subcellular distributions: short Osk is located on polar granules, ribonucleoprotein complexes in the germ plasm, whereas long Osk is present on the endosomal surface [10].

In vitellogenic stage oocytes, the posterior pole region where the germ plasm is assembled shows much higher endocytic activity and enriches more endosomes at least until stage 10b when ooplasmic streaming begins [10,11]. Endocytic regulation in the oocyte is involved in the maintenance of proper microtubule alignments along which *osk* mRNA is transported. In oocytes defective in the endocytic pathway, such as hypomorphic *rab11* and *rabenosyn-5* (*rbsn-5*)-null mutants, posterior enrichment of microtubule plus ends fails to be maintained, which leads to mislocalization of *osk* mRNA [11–13].

The endocytic activity also acts downstream of long Osk in anchoring germ plasm components to the cortex. The local activation of endocytosis in the oocyte depends on *osk*. In *osk* mutant oocytes, endocytosis is not locally activated but occurs at a low level along the entire cortex [10,11]. In contrast, misexpression of long Osk, but not short Osk, at the oocyte anterior induces ectopic activation of endocytosis [11]. Long Osk-induced endocytosis promotes the formation of long F-actin fibers that emanate from cortical F-actin bundles [10]. These long F-actin projections are also induced where Osk, particularly long Osk, is ectopically expressed [11,14]. However, in endocytosis-defective oocytes such as *rab5* and *rbsn-5* germline clones, ectopic Osk leads to formation of aberrant F-actin aggregates, and diffusion of germ plasm components from the oocyte cortex [11,15]. We have previously shown that a conserved endosomal/Golgi protein, Mon2, acts together with actin nucleators, Cappuccino (Capu) and Spire (Spir), in long Osk-mediated actin remodeling [15]. This finding suggests that endocytic vesicles formed by long Osk-driven endocytosis might provide platforms for actin remodeling. However, it remains unknown whether a specific set of receptor-ligand combinations acts in endocytosis-dependent actin remodeling and germ plasm anchoring.

In this work, we found that long Osk associated with Yolkless (Yl), which mediates incorporation of nutrient yolk proteins into the oocyte. In mutant oocytes for *yl* or its ligands [Yolk protein 1–3 (Yp1–3) and newly identified Apolipophorin fragments], the posterior enrichment of microtubule plus-ends failed to be maintained, which led to *osk* mRNA mislocalization. We also showed that these yolk uptake-defective oocytes did not properly respond to long Osk, resulting in actin disorganization and diffusion of germ plasm components from the cortex. Therefore, ligand-dependent endocytosis of Yl is not merely the nutrition uptake process for future embryonic development, but also crucial for *osk* mRNA localization and cortical anchorage of germ plasm components in the oocyte during oogenesis.

## Results

### Association of long Osk with yolk protein receptor Yolkless

Since long Osk is located on the endosomal surface [10], we proposed that it may contribute to actin remodeling through interaction with proteins that also reside on endosomes. We

conducted a proteomic approach to identify potential factors that act together with long Osk in actin remodeling and germ plasm anchoring. To obtain sufficient Osk expression for proteomic analyses, we used transgenic flies overexpressing either the *3×FLAG-GFP*-tagged *long* or *short osk* coding sequence fused with the *bicoid* (*bcd*) *3′ UTR*, which leads transcripts to the anterior pole of the oocyte. Anteriorly expressed Osk directs ectopic assembly of the germ plasm [16]. We confirmed that anteriorly expressed 3×FLAG-GFP-long and short Osk, like untagged isoforms, recruited endosomal proteins such as Rab11 and germ plasm components such as Vas, respectively (S1A and S1B Fig). Long Osk also recruited a trace amount of Vas (S1A Fig), likely due to its subtle activity to induce germ plasm assembly [5]. The anti-FLAG immunoprecipitates from ovarian extracts of control and these transgenic females were resolved by SDS-PAGE and were visualized by silver staining (Fig 1A and S1C Fig). Mass spectrometric analyses of reproducibly coprecipitated proteins revealed that several proteins known to associate with RNA-containing complexes commonly coprecipitated with both Osk isoforms (Fig 1A, S1C Fig, and S1 Table) [17–19]. Given that Osk interacts directly with RNAs [20,21], it might associate with these proteins via RNAs. We also found that the approximately 250 kDa protein, which preferentially coimmunoprecipitated with long Osk (Fig 1A), was Yl.

Yl is a member of the low-density lipoprotein receptor (LDLR) family and is specifically expressed in female germline during oogenesis [22–28] (S2A Fig). Yl mediates uptake of yolk proteins, which are produced by somatic cells, into the oocyte from stage 8 onwards. Endocytosed Yl is recycled back to the oocyte surface, whereas yolk proteins are directed to lysosome-related organelles, called yolk granules, which store nutrition for future embryonic development.

Western blot analysis showed a strong Yl signal in anti-FLAG immunoprecipitates from ovarian lysates overexpressing 3×FLAG-GFP-long Osk (Fig 1B). In contrast, only a trace amount of Yl was coimmunoprecipitated with short Osk. Furthermore, endosomal markers such as Rab5, Rab7, and Rab11 did not coimmunoprecipitate with long Osk (Fig 1B). Thus, our immunoprecipitation conditions were sufficiently stringent to identify long Osk-associated proteins while generally excluding other endosomal components. We next examined colocalization of 2 Osk isoforms with Yl in the posterior region of the stage 10b oocyte using transgenic flies expressing either 3×FLAG-GFP-tagged long or short Osk isoform under the *osk* promoter and *3′ UTR*. Colocalization analyses using high magnified and deconvoluted images revealed that, while the signals for long Osk were faint on the plasma membrane where the Yl signals were high, they were highly overlapped with internalized Yl in the ooplasmic area (Fig 1C). In contrast, short Osk was present in the more interior area in the ooplasm (Fig 1D). The Pearson correlation coefficient between long Osk and Yl was significantly higher than that of short Osk and Yl (Fig 1E and S1 Data). This is consistent with the previous report that, under immunoelectron microscopy, long Osk is detected on the endosomes whereas short Osk on the polar granules [10]. Furthermore, when Yl and Osk were coexpressed in *Drosophila* S2 cells, long Osk was colocalized with Yl on vesicular structures in the cytoplasm, whereas short Osk accumulated in the nucleus as reported [29,30] and was not colocalized with Yl (S1D and S1E Fig). These results suggest that long Osk specifically interacts with Yl on the endosomal membrane.

## Generation of null mutants for *yl* and its ligand encoding genes

Several *yl* alleles have been reported [26], but their precise molecular lesions are unknown. We therefore generated molecularly definitive *yl*-null mutants. Using CRISPR-Cas9 technology, we obtained 2 frameshift alleles, $yl^{16-2}$ and $yl^{16-6}$ (S2B Fig). Yl proteins were undetectable in

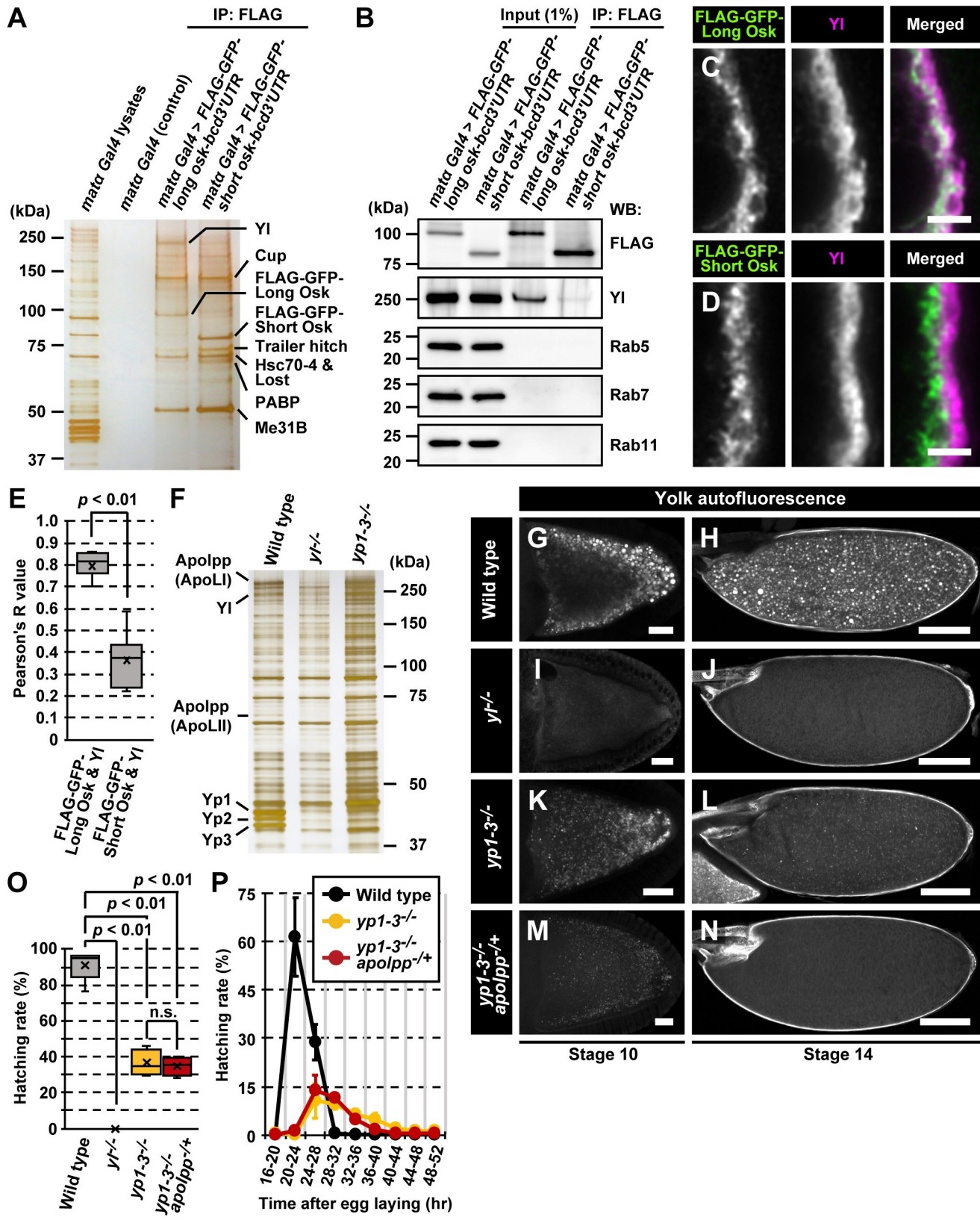

**Fig 1. Long Osk associates with the Yolk protein receptor, Yl.** (A) An image of NuPAGE gel stained by silver. Lysates were prepared from ovaries overexpressing ectopically 3×FLAG-GFP-long or short Osk, which was driven by matα-Gal4, at the anterior pole of the oocyte. Ovarian lysates were immune-affinity purified with anti-FLAG beads, and precipitants were separated and were silver stained. Reproducibly precipitated bands were analyzed by mass spectrometry for protein identification. (B) Ovarian lysates expressing 3×FLAG-GFP-long or short Osk were immunoprecipitated with anti-FLAG antibodies and were analyzed by western blots using antibodies against FLAG, Yl, Rab5, Rab7, and Rab11. (C, D) Localization of FLAG-GFP-tagged Osk isoforms and Yl at the posterior pole of stage 10b oocytes. The posterior pole of oocytes is oriented to the right throughout the figures. (E) Pearson correlation coefficient values between 3×FLAG-GFP-tagged Osk isoforms and Yl. The Pearson values were calculated from 10 randomly selected stage 10b oocytes in each group. The $p$-value was calculated by Mann–Whitney $u$ test. (F) Silver staining of ovarian lysates from wild-type, $yl^{-/-}$, and $yp1$–$3^{-/-}$ females. In addition to 3 bands corresponding to Yp1–3 at about 45 kDa, 2 bands at about 300 and 70 kDa were absent in $yl$-deficient extracts. They were identified as Apolpp fragments (ApoLI and ApoLII). (G–N) Yolk autofluorescent granules in stage 10 (G, I, K, M) and stage 14 oocytes (H, J, L, N) of wild type (G, H), $yl^{-/-}$ (I, J), $yp1$–$3^{-/-}$ (K, L), and $yp1$–$3^{-/-}$ $apolpp^{-/+}$ (M, N). Yolk autofluorescence was absent in $yl^{-/-}$ oocytes. In contrast, autofluorescence was still detected in $yp1$–$3^{-/-}$ or $yp1$–$3^{-/-}$ $apolpp^{-/+}$ oocytes and remained at a very low level at stage 14. Note that stage 14 $yl$-null or Yl-ligand-depleted oocytes were morphologically normal. (O) Hatching rates of eggs laid by females of wild-type ($y\ w$), $yl^{-/-}$, $yp1$–$3^{-/-}$, and $yp1$–$3^{-/-}$ $apolpp^{-/+}$ mutants. Complete genotypes were: $yl^{-/-}$: $yl^{16-2/16-6}$, $yp1$–$3^{-/-}$: $yp1$–$3^{7/111}$, $yp1$–$3^{-/-}$ $apolpp^{-/+}$: $yp1$–$3^{7/111}$ $apolpp^{39\ or\ 102}/Ci^D$. Numbers of hatched or unhatched eggs were counted 3 days after egg deposition. $p$-values were calculated by Mann–Whitney $u$ test. (P) Hatching rates of eggs laid by wild-type ($y\ w$), $yp1$–$3^{-/-}$, and $yp1$–$3^{-/-}$ $apolpp^{-/+}$ females at consecutive time points after egg laying. Eggs were laid for 4 h, and hatching rate was plotted every 4 h. Standard deviations are indicated by error bars. Uncropped gel and blot images for panels A, B, and F can be found in S1 Raw Images. Numerical data for panels E, O, and P can be found in S1 Data. Scale bars: 2 μm (C, D), 20 μm (G, I, K, M), or 100 μm (H, J, L, N). GFP, green fluorescent protein; IP, immunoprecipitation; n.s, not significant; Osk, Oskar; Yl, Yolkless.

ovarian extracts from these mutant females (S2C Fig). Three major bands corresponding to Yolk protein 1–3 (46, 45, and 44 kDa for Yp1, Yp2, and Yp3, respectively) [31] were also absent in ovarian proteins of the $yl^{16-2}/yl^{16-6}$ females (Fig 1F). In addition, vitellogenic (stage 10) and matured (stage 14) oocytes of $yl^{16-2}/yl^{16-6}$ females were entirely devoid of autofluorescence of yolk granules, which normally fill the ooplasm (Fig 1G–1J). Although $yl^{16-2}/yl^{16-6}$ oocytes completed oogenesis without morphological abnormalities (Fig 1J), eggs laid by $yl^{16-2}/yl^{16-6}$ females were shriveled and never hatched (Fig 1O and S1 Data). Thus, Yl is required for rehydration, a potential trigger for egg activation, during ovulation of fully developed oocytes; yet the lack of yolk uptake does not globally affect oogenesis. For further analyses, we used the $yl^{16-2}/yl^{16-6}$ allelic combination as the $yl$-null mutant.

Silver-stained SDS-PAGE gels of ovarian extracts showed that, in addition to Yp1–3, two additional bands of about 300 and 70 kDa were absent in $yl$-null mutants (Fig 1F). By mass spectrometric analysis, they were identified as Apolipophorin (Apolpp; also known as Retinoid- and fatty acid-binding protein). Apolpp is a vitellogenin-like lipoprotein and a homologue to vertebrate Apolipoprotein B (ApoB) [32,33]. $apolpp$ encodes a large approximately 400 kDa protein, which is proteolytically processed to generate approximately 70 and 300 kDa fragments (ApoLII and ApoLI for N- and C-terminal parts, respectively) [33]. The $apolpp$ transcript is detected at high levels in many somatic tissues but is almost undetectable in ovaries [34] (http://flybase.org). Therefore, it is likely that Apolpp fragments secreted from somatic cells are incorporated into the oocyte through Yl-mediated endocytosis.

Apolpp is known to transport various types of lipids into hemolymph [32,33,35]. However, there was no report that Apolpp is a ligand for Yl or behaves as a yolk protein. We generated compound mutants for $yp1$, $yp2$, $yp3$, and $apolpp$ using CRISPR-Cas9 technology (S2D and S2E Fig), and found that $yp1$–$3$ triple null-mutants (hereafter denoted as $yp1$–$3^{-/-}$) were viable, whereas $apolpp$-null mutants were homozygous lethal as reported [33].

It has been reported that eggs laid by females with reduced copy number of $yp1$–$3$ genes frequently do not hatch [36]. It suggests that protein products from the 3 genes are essential for embryonic development. Unexpectedly, we observed that approximately 35% eggs laid by $yp1$–$3^{-/-}$ females hatched and developed into fertile adults despite a significant increase in time for embryogenesis (Fig 1O and 1P and S1 Data). Furthermore, despite the lack of Yp1–3 bands in $yp1$–$3^{-/-}$ ovarian extracts (Fig 1F), autofluorescent granules in vitellogenic and mature $yp1$–$3^{-/-}$ oocytes, while they were fewer and smaller, were still detected (Fig 1K and 1L). This suggested that Apolpp fragments endocytosed through Yl contributed to forming

yolk granules and exhibited autofluorescence. To test this, we removed 1 copy of *apolpp* in the *yp1–3*$^{-/-}$ background. The autofluorescence in *yp1–3*$^{-/-}$ *apolpp*$^{-/+}$ oocytes was further reduced compared with *yp1–3*$^{-/-}$ oocytes (Fig 1M and 1N), although *yp1–3*$^{-/-}$ *apolpp*$^{-/+}$ females still laid eggs with hatching ability (Fig 1O and 1P and S1 Data). These results indicate that cleaved Apolpp fragments are hitherto unidentified yolk proteins that are endocytosed as Yl-ligands. In the following experiments, we used the *yp1–3*$^{-/-}$ *apolpp*$^{-/+}$ females that produce Yl ligand-depleted oocytes to address ligand-dependent roles of Yl.

## Yl and its ligands are crucial for localization of germ plasm components to the oocyte posterior cortex

Because long Osk is involved in anchoring the germ plasm to the oocyte cortex, Yl and its ligands may also contribute to this process. Therefore, we first examined localization of a germ plasm marker, YFP-Vas, to the posterior pole of the oocyte. In wild-type oocytes, YFP-Vas was first detectable at the posterior end at stage 9, and remained tightly localized, with gradually increasing accumulation, until completion of oogenesis (stage 14) (Fig 2A and 2D). However, *yl*$^{-/-}$ oocytes showed diffused localization of YFP-Vas in the posterior ooplasm at stage 9 (Fig 2B). In stage 14 *yl*$^{-/-}$ oocytes, YFP signals at the posterior were very faint or even undetectable (Fig 2B and 2D). Similarly, hemizygotes for *yl*-null (*yl*$^{16-2}$/*Df(1)KA9* or *yl*$^{16-6}$/*Df(1)KA9*) showed defects in the posterior localization of YFP-Vas (S3 Fig). This result supports the idea that the loss of *yl* caused the phenotype. YFP-Vas localization was also aberrant in Yl ligand-depleted oocytes (Fig 2C). The YFP signals at the posterior region of Yl ligand-depleted stage 14 oocytes were reduced to about 60% of the wild type level but were significantly higher than those in *yl*$^{-/-}$ oocytes (Fig 2D and S1 Data). This difference might be caused by the residual Apolpp in Yl ligand-depleted oocytes. These results nevertheless indicate that the proper germ plasm accumulation at the oocyte posterior cortex requires Yl and its ligands. Consistently, we found that no eggs laid by *yl*$^{-/-}$ females proceeded to embryogenesis, but embryos derived from Yl ligand-depleted females had approximately 25% fewer pole cells than wild-type embryos (Fig 2E–2G and S1 Data).

Vas is recruited to the future germ plasm region through interaction with short Osk, which is produced from localization-coupled translation of *osk* mRNA [7,20]. We therefore examined localization of Staufen (Stau), a marker for *osk* mRNA [37]. In wild-type oocytes, Stau was tightly localized to the posterior pole at stages 9 and 10b (Fig 2H and 2K). We found that, in *yl*$^{-/-}$ oocytes, Stau was not thoroughly targeted to the posterior pole and was observed distant from the posterior cortex (Fig 2I and 2L). Yl ligand-depleted oocytes also showed diffused localization of Stau, although the defects were somewhat moderate (Fig 2J and 2M). Quantitative intensity plots showed that Stau was distributed to more interior regions in *yl*$^{-/-}$ and Yl ligand-depleted oocytes (Fig 2N and S1 Data). We also confirmed that *osk* mRNA was not properly localized at the posterior pole in *yl*$^{-/-}$ or Yl ligand-depleted oocytes (S4 Fig). Consistent with the mislocalization of *osk* mRNA, Osk protein was not tightly localized to the cortex but was diffused towards the ooplasm in *yl*$^{-/-}$ or Yl ligand-depleted oocytes (Fig 2O–2U and S1 Data). These results suggest that the reduced accumulation of germ plasm components in the yolk uptake-defective oocytes was, at least in part, attributable to mislocalization of *osk* mRNA.

## Yl and its ligands are crucial for the maintenance of polarized microtubule alignment

The posterior localization of *osk* mRNA depends on the microtubule alignment along the anterior-posterior axis in the oocyte, where the plus ends of microtubule arrays are weakly

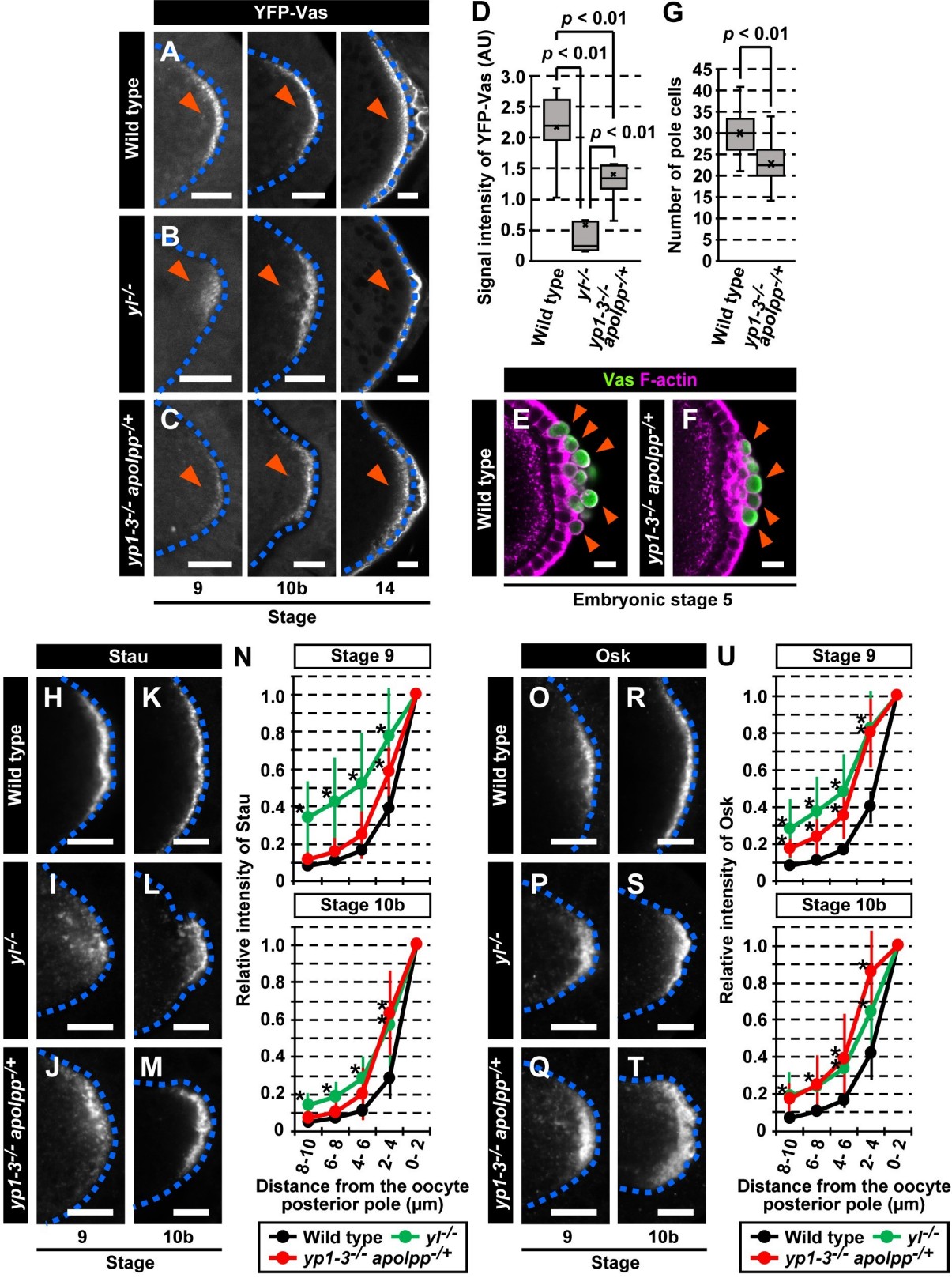

**Fig 2. Yl and its ligands are required for posterior localization of germ plasm components in oocytes.** (A–C) Localization of YFP-Vas at the posterior region of oocytes in wild type (A), $yl^{-/-}$ (B), and $yp1–3^{-/-}$ $apolpp^{-/+}$ (C) ovaries at stage 9, 10b, and 14. Arrowheads indicate the posterior pole. Posterior accumulation of YFP-Vas was defective in $yl^{-/-}$ and $yp1–3^{-/-}$ $apolpp^{-/+}$ oocytes. (D) Signal intensity of YFP-Vas at the posterior region of stage 14 oocytes in wild type ($n = 7$), $yl^{-/-}$ ($n = 8$), and $yp1–3^{-/-}$ $apolpp^{-/+}$ ($n = 8$). $p$-values were calculated by Mann–Whitney $u$ test. (E, F) Pole cells in stage 5 embryos derived from wild-type (E) and $yp1–3^{-/-}$ $apolpp^{-/+}$ (F) females were stained for Vas. Vitelline membrane of fixed embryos was hand peeled without methanol treatment, so that F-actin-rich cell boundaries could be visualized with Phalloidin (magenta). Arrowheads point to pole cells that were positive for Vas. (G) Number of Vas-positive pole cells in stage 5 progeny embryos of wild-type and $yp1–3^{-/-}$ $apolpp^{-/+}$ females. Twenty embryos were counted for each genotype. The $p$-value was calculated by Mann–Whitney $u$ test. (H–M) Posterior pole of stage 9 and 10b oocytes in wild type (H, K), $yl^{-/-}$ (I, L), and $yp1–3^{-/-}$ $apolpp^{-/+}$ (J, M) ovaries stained for Stau. (N) Relative intensity plots for Stau at the posterior region of oocytes in wild type ($n = 11$ at stage 9 and $n = 10$ at stage 10b), $yl^{-/-}$ ($n = 11$ at stage 9 and $n = 11$ at stage 10b), and $yp1–3^{-/-}$ $apolpp^{-/+}$ ovaries ($n = 13$ at stage 9 and $n = 12$ at stage 10b). The intensities are defined as 1.0 at 0–2 μm distance from the posterior cortex in each genotype and are plotted every 2 μm along the posterior to the anterior axis. Standard deviations are indicated by error bars. Asterisks above the markers indicate significant differences from wild type controls at $p < 0.01$. $p$-values were calculated by Mann–Whitney $u$ test. (O–T) Posterior pole of stage 9 and 10b oocytes in wild type (O, R), $yl^{-/-}$ (P, S), and $yp1–3^{-/-}$ $apolpp^{-/+}$ (Q, T) ovaries stained for Osk. (U) Relative intensity plots of Osk at the posterior region of oocytes in wild type ($n = 11$ at stage 9 and $n = 10$ at stage 10b), $yl^{-/-}$ ($n = 9$ at stage 9 and $n = 12$ at stage 10b), and $yp1–3^{-/-}$ $apolpp^{-/+}$ ovaries ($n = 13$ at stage 9 and $n = 12$ at stage 10b). The intensities are defined as 1.0 at 0–2 μm distance from the posterior cortex in each genotype and are plotted every 2 μm along the posterior to the anterior axis. Standard deviations are indicated by error bars. Asterisks above the markers indicate significant differences from wild-type controls at $p < 0.01$. $p$-values were calculated by Mann–Whitney $u$ test. Oocytes are outlined by blue dashed lines. Numerical data for panels D, G, N, and U can be found in S1 Data. Scale bars: 10 μm. Osk, Oskar; Stau, Staufen; Vas, Vasa; YFP, yellow fluorescent protein; Yl, Yolkless.

enriched at the posterior pole [3,4]. We thus examined localization of a microtubule plus-end marker, Kinesin-β-galactosidase (Kin-βgal) [3]. In wild-type oocytes, Kin-βgal becomes enriched to the posterior pole at stage 8 and forms a crescent at the posterior cortex at stage 9 (Fig 3A and 3D). In stage 8 $yl^{-/-}$ or Yl ligand-depleted oocytes, Kin-βgal was concentrated to the oocyte posterior as in wild type (Fig 3A–3C). However, in the subsequent stage 9 oocytes, Kin-βgal was not tightly localized to the posterior cortex but was diffused into the ooplasm (Fig 3E and 3F). Stau also failed to be targeted to the posterior pole of these oocytes. These results indicate that Yl and its ligands are dispensable for the initial establishment, but are required for the maintenance, of plus end targeting of microtubules toward the posterior pole of the oocyte.

We also used a Nod-βgal fusion protein that leads βgal to the anterior, where the minus ends of microtubules are highly enriched (arrowheads in Fig 3G) [38]. In $yl^{-/-}$ or Yl ligand-depleted oocytes, Nod-βgal was normally localized at the anterior corners as in wild type (arrowheads in Fig 3H and 3I). These results indicate that the microtubule organizing centers were normally distributed even in the absence of Yl or by the depletion of Yl ligands. In addition, we found that, in $yl^{-/-}$ and Yl ligand-depleted oocytes, *bcd* and *gurken* (*grk*) mRNAs, whose localizations are dependent on the microtubule minus end-directed motor, Dynein [39], were normally localized at the anterior pole and anterior-dorsal region, respectively (Fig 3J–3L and Fig 3P–3R). Their localization remained unaffected even at later stages (Fig 3M–3O and Fig 3S–3U). Thus, we conclude that Yl and its ligands are specifically required for *osk* mRNA localization to the posterior pole of the oocyte.

## Long Osk can stimulate endocytosis even in the absence of Yl or by the depletion of its ligands

We next examined roles of Yl in endocytic stimulation driven by long Osk. Wild-type oocytes incorporate an endocytic tracer dye, FM4-64, from the entire cortex, with preferential uptake from the posterior pole where Osk resides (Fig 4A) [10,11]. We found that the loss of Yl or the depletion of Yl ligands led to reduction of FM4-64 signals at the lateral and posterior regions in the oocyte (Fig 4B and 4C). These results indicate that the yolk uptake contributes to the major endocytic activity in vitellogenic stage oocytes as reported [24]. However, *yl*-null or Yl ligand-depleted oocytes still showed enrichment of FM4-64 signals at the posterior region

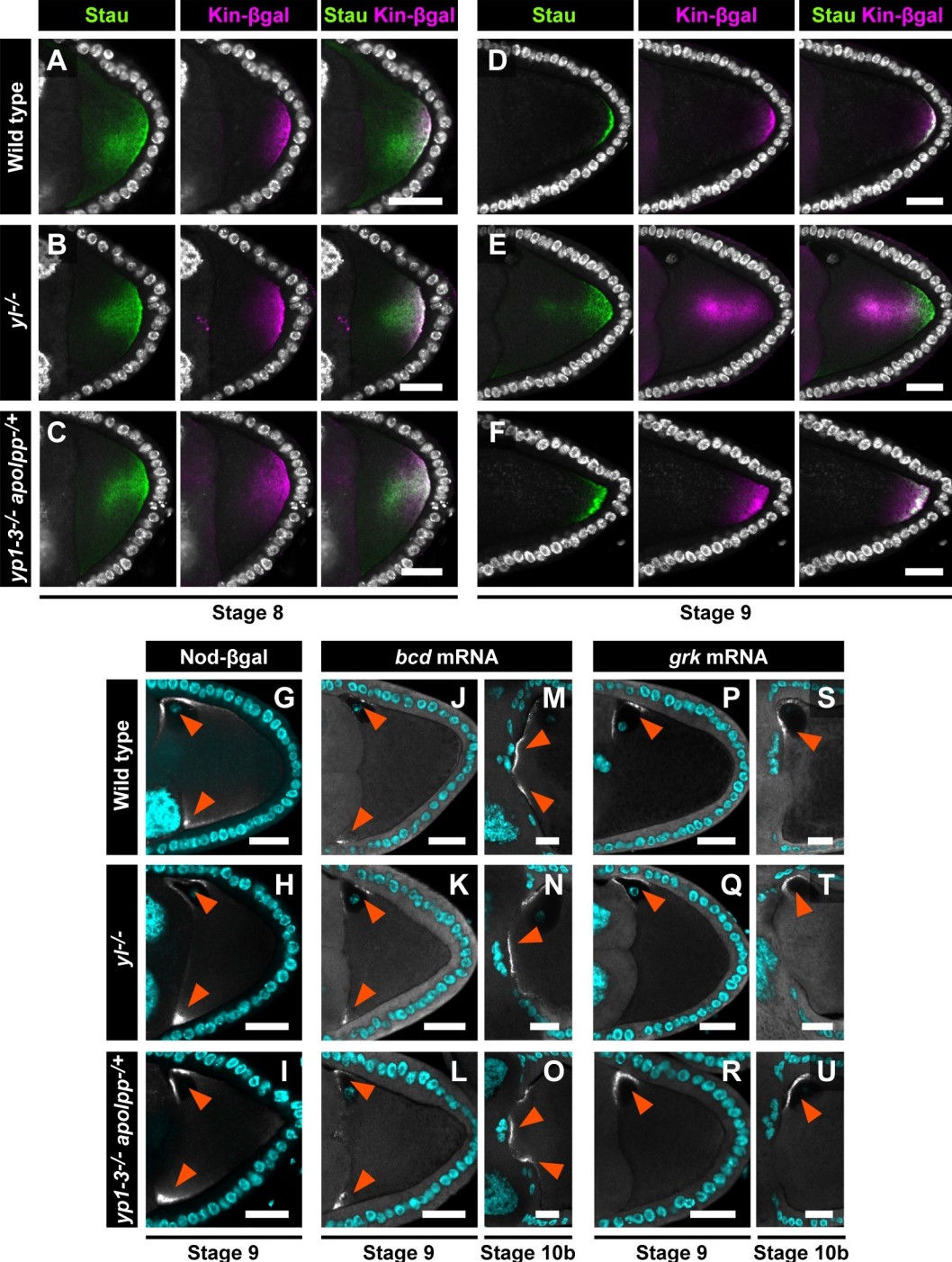

**Fig 3. Yl and its ligands are required for microtubule-dependent localization of *osk* mRNA to the oocyte posterior cortex.** (A–F) Localization of Stau and Kin-βgal in stage 8 and stage 9 oocytes in wild-type (A, D), *yl*−/− (B, E), and *yp1–3*−/− *apolpp*−/+ (C, F) ovaries. DNA stained with DPAI is shown in white. The microtubule plus-end marker Kin-βgal was accumulated at the posterior pole in *yl*−/− and *yp1–3*−/− *apolpp*−/+ oocytes at stage 8 (A–C). However, its localization to the posterior pole was not maintained in stage 9 *yl*−/− and *yp1–3*−/− *apolpp*−/+ oocytes (D–F). Localization of Stau was also disrupted. (G–I) Localization of Nod-βgal in stage 9 oocytes in wild-type (G), *yl*−/− (H), and *yp1–3*−/− *apolpp*−/+ (I) ovaries. DNA stained with DPAI is shown in cyan. (J–U) Localization of *bcd* (J–O) and *grk* mRNAs (P–U) in stage 9 or stage 10b oocytes in wild-type (J, M, P, S), *yl*−/− (K, N, Q, T), and *yp1–3*−/− *apolpp*−/+ (L, O, R, U) ovaries. DNA stained with DPAI is shown in cyan. Localization of *bcd* and *grk* mRNA was intact in *yl*−/− and *yp1–3*−/− *apolpp*−/+ oocytes (arrowheads). Scale bars: 20 μm. Kin-βgal, Kinesin-β-galactosidase; Stau, Staufen; Yl, Yolkless.

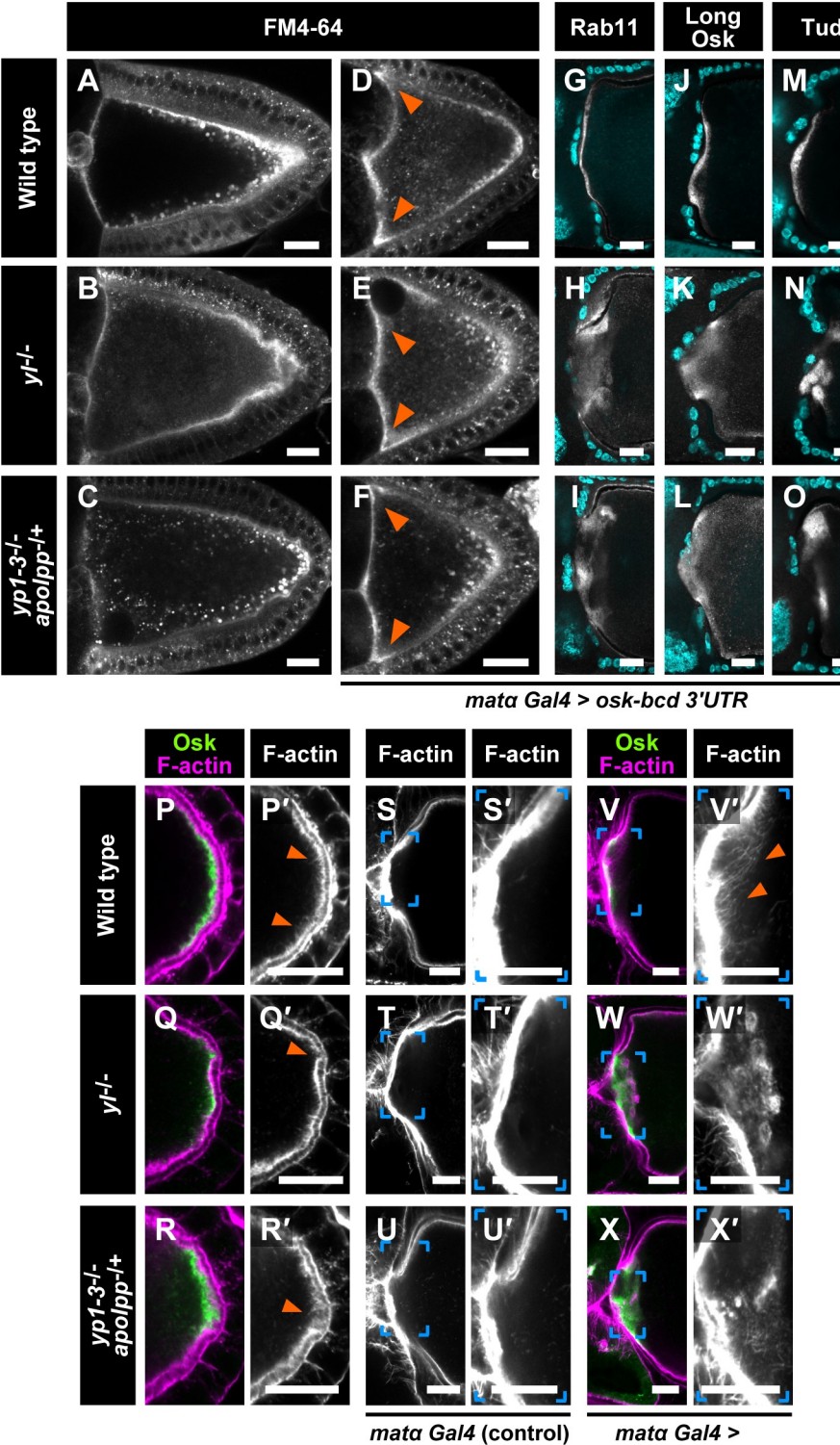

**Fig 4. Osk-mediated actin remodeling and cortical anchorage of germ plasm components depend on Yl and its ligands.** (A–C) Internalized FM4-64 dye in stage 10a oocytes. Wild-type oocytes showed strong FM4-64 signals concentrated at the posterior pole (A). In contrast, $yl^{-/-}$ (B) and $yp1$–$3^{-/-}$ $apolpp^{-/+}$ (C) oocytes displayed reduced FM4-64 uptake at the lateral and posterior regions. However, the posterior enrichment of FM4-64 signals remained in these mutant oocytes. Note that in $yl^{-/-}$ oocytes, the signals appeared as smaller vesicular compartments. (D–F) Stage 9 $osk$-$bcd$ $3'UTR$ oocytes cultured in the presence of FM4-64. Arrowheads indicate ectopic FM4-64 uptake in these

oocytes. (G–O) Signals for Rab11 (G–I), long Osk (J–L), and Tud (M–O) at the anterior region of the oocytes expressing *osk-bcd 3′ UTR* in wild-type (G, J, M), $yl^{-/-}$ (H, K, N), and $yp1-3^{-/-}$ $apolpp^{-/+}$ ovaries (I, L, O). DNA stained with DPAI is shown in cyan. (P–R) The posterior region of stage 10b oocytes stained for Osk and F-actin in wild-type (P), $yl^{-/-}$ (Q), and $yp1-3^{-/-}$ $apolpp^{-/+}$ (R) ovaries. Arrowheads in P′ indicate posterior F-actin projections in wild-type oocytes. Arrowheads in Q′ and R′ point to disorganized cortical F-actin bundles. (S–U) The anterior region of stage 10b oocytes stained for F-actin in wild-type (S), $yl^{-/-}$ (T), and $yp1-3^{-/-}$ $apolpp^{-/+}$ mutants (U). Right panels (S′–U′) show high-magnification views of the bracketed area in the left panels. In the absence of ectopic Osk, the cortical F-actin layer at the anterior region was intact in $yl^{-/-}$ and $yp1-3^{-/-}$ $apolpp^{-/+}$ mutant oocytes. (V–X) The anterior region of stage 10b *osk-bcd 3′UTR* oocytes stained for Osk and F-actin. Right panels (V′–X′) show high-magnification views of the bracketed area in the left panels. Ectopic Osk induced long F-actin projections in the wild-type oocyte (arrowheads in V′). However, it induced aberrant F-actin aggregates in $yl^{-/-}$ and $yp1-3^{-/-}$ $apolpp^{-/+}$ oocytes (W′ and X′). Scale bars: 20 μm. Osk, Oskar; Tud, Tudor; Yl, Yolkless.

(Fig 4B and 4C). This suggests that Osk retains its activity to stimulate endocytosis. To further examine this possibility, we analyzed effects of ectopic Osk expression at the anterior pole of the $yl^{-/-}$ and Yl ligand-depleted oocytes. In *osk-bcd 3′UTR* oocytes, enhanced FM4-64 uptake and recruitment of endosomal proteins such as Rab5 and Rab11 at the anterior pole were observed (Fig 4D and 4G and S5A, S5D, and S5G Fig) [11]. When Osk was misexpressed at the oocyte anterior pole in $yl^{-/-}$ or Yl ligand-depleted oocytes, ectopic FM4-64 uptake occurred as in wild-type oocytes (Fig 4E and 4F). Endosomal proteins such as Rab5 and Rab11 were also recruited to the anterior region (Fig 4H and 4I and S5B, S5C, S5E, S5F, S5H, and S5I Fig). These results indicate that, even in the absence of Yl or by the depletion of Yl ligands, endocytosis can be locally activated in response to Osk.

## Yl and its ligands are crucial for long Osk-mediated actin remodeling and cortical anchorage of germ plasm components

In $yl^{-/-}$ and Yl ligand-depleted oocytes expressing the *osk-bcd 3′UTR* transgene, ectopic long Osk and endosomal proteins were not localized to the cortex but diffused into the ooplasm (Fig 4G–4L and S5E and S5F Fig). Because localization of *bcd* mRNA was unaffected in these mutant oocytes (Fig 3J–3O), ectopic long Osk that was produced at the anterior cortex appeared not to be anchored. In addition, another germ plasm protein Tud, which is recruited to the anterior pole by ectopic Osk expression [40], was not tightly localized to the oocyte cortex (Fig 4M–4O). These results indicate that Yl-mediated endocytosis acts downstream of long Osk in anchoring germ plasm components to the oocyte cortex.

Long Osk-mediated endocytic activation promotes actin remodeling to form long F-actin projections, which are thought to anchor germ plasm components to the oocyte cortex [10,11]. This Osk-mediated actin remodeling requires proper regulation of the endocytic pathway. For example, *rbsn-5*-null oocytes do not properly respond to Osk for the formation of F-actin projections [11], and *rab5*-null oocytes show a disrupted cortical F-actin layer, which is suppressed by the simultaneous loss of *osk* [15]. We thus examined the effects of the loss of Yl or the depletion of Yl ligands on long Osk-dependent actin remodeling. As reported [10,11], long F-actin projections were emerged from the posterior cortical actin bundles in wild-type oocytes (Fig 4P and 4P', arrowheads). We found that, in both $yl^{-/-}$ and Yl ligand-depleted oocytes, cortical F-actin was disorganized at the posterior pole, particularly at the regions where Osk was highly enriched (Fig 4Q, 4Q', 4R, and 4R', arrowheads).

Ectopic long F-actin projections were induced at the anterior pole of the oocyte when Osk was misexpressed (Fig 4S, 4S', 4V, and 4V', arrowheads) [11,14]. In these oocytes, Osk was tightly associated with the oocyte cortex (Fig 4V). However, the ectopic Osk in $yl^{-/-}$ oocytes caused many aberrant F-actin aggregates (Fig 4T, 4T', 4W, and 4W'). Actin aggregation was

also induced in Yl ligand-depleted oocytes expressing ectopic Osk (Fig 4U, 4U', 4X, and 4X'). These results indicate that the ligand-dependent endocytosis of Yl is required for long Osk-mediated actin remodeling to anchor germ plasm components to the oocyte cortex.

## Discussion

In egg-laying animals, the principal nutrient source for embryonic development is yolk proteins, which accumulate in oocytes during oogenesis. Yolk precursors in most organisms are not synthesized by the oocyte but are produced by somatic cells and are incorporated into the oocyte through receptor-mediated endocytosis [41]. In *Drosophila*, yolk proteins, Yp1–3, are predominantly synthesized in fat bodies and ovarian follicle cells [42–44]. Yp1–3 are secreted from these tissues and are selectively taken up in the developing oocyte by the yolk receptor, Yl, which binds yolk proteins on the oocyte surface and mediates their internalization by clathrin-mediated endocytosis. It has been well known that Yl-mediated endocytosis is essential for incorporation and storage of nutrient yolk proteins. We now show that the process is also crucial for the maintenance of polarized microtubule arrays that promote *osk* mRNA localization, and long Osk-mediated actin remodeling that leads to the anchoring of germ plasm components to the oocyte cortex (S6 Fig).

In addition to Yp1–3 proteins, we have identified processed Apolpp fragments (ApoLI and ApoLII) as Yl-ligand proteins (Fig 1). Yolk proteins can be classified into 2 distinct groups [45]. The major group is vitellogenins, which are found in most oviparous animals such as fish, frog, chicken, and nematode. The other group is called yolk proteins and is found specifically in dipteran insects including *Drosophila*. Despite the similarity in their physiological roles, both groups are heterogeneous in their primary sequences. Given that Apolpp is a vitellogenin group protein, it is conceivable that dipteran insects also utilize vitellogenins as yolk proteins. In somatic cells, *Drosophila* Apolpp has vital functions in morphogenesis; it forms complexes with lipid-linked signaling molecules, such as Hedgehog and Wingless, and helps to establish their proper morphogen gradient in somatic tissues such as wing discs [35,46]. Interestingly, RNAi-mediated *apolpp* knockdown in the fat body of the adult female causes degeneration of egg chambers during previtellogenic stages of oogenesis [47]. Thus, Apolpp appears to have, in addition to functions as yolk proteins, vital roles in oogenesis independently of Yl.

Yolk-depleted eggs from $yp1-3^{-/-} apolpp^{-/+}$ females frequently completed embryogenesis and developed into adults (Fig 1O and S1 Data). These findings indicate that enormous amounts of yolk proteins are not necessarily critical for successful embryonic development. Similarly, in *Caenorhabditis elegans* eggs, yolk is largely dispensable for embryogenesis [48,49], but its titer impacts on postembryonic phenotypes such as developmental speed, starvation resistance, and fecundity [50]. In addition, we found that the yolk depletion caused reduction in the number of pole cells in stage-5 embryos (Fig 2G and S1 Data). Thus, there might be a mechanism for adjusting fecundity to food availability.

Yl was associated with long Osk presumably on endocytic vesicles (Fig 1). Yl is a single-pass transmembrane protein with a large extracellular domain that binds to yolk proteins [28] (S2A Fig). The cytoplasmic tail of Yl contains 2 endocytic sorting signals: the noncanonical FXNPXA sorting sequence and the atypical dileucine sequence, which are recognized by Ced-6 and the AP-2 complex, respectively [27]. Ced-6 and the AP-2 complex act as clathrin adaptors and redundantly promote endocytosis of Yl [27]. Given that long Osk is produced in the cytoplasm, it would associate with the cytoplasmic tail of Yl. It is an interesting future issue to examine whether the endocytic sorting signals of Yl and/or their binding adaptors are involved in association and function of long Osk.

We showed that Yl-mediated endocytosis was required for actin remodeling to form long actin projections that are proposed to anchor germ plasm components to the oocyte cortex (Fig 4). In oocytes defective in Yl-mediated endocytosis, cortical actin at the posterior region was disorganized, and ectopic Osk at the anterior pole induced aberrant F-actin aggregates. Similar defects have been observed in endocytosis-defective *rab5* or *rbsn-5* oocytes [11,15]. Notably, in $yl^{-/-}$ or Yl ligand-depleted oocytes, endocytosis can be activated in response to Osk (Fig 4). Thus, it is likely that aberrant actin remodeling in these mutant oocytes is caused specifically by the absence of long Osk-Yl association, rather than by a general decrease in endocytic activity. Our findings further suggest that an amount of Yl on endosomes might be critical for the proper actin remodeling to anchor germ plasm components to the oocyte cortex.

We previously identified several actin regulators (Mon2, Capu, and Spir) that are involved in Osk-mediated actin remodeling [15]. These actin regulators appear to be localized on membranous structures, such as endosomes. Mon2 is localized on Rab7- or Rab11-positive vesicles as well as Golgi in *Drosophila* and mammalian cells [15,51]. In addition, in mouse oocytes, proteins homologous to Spir (Spire1 and Spire2) and Capu (Formin-2) are colocalized on Rab11a-positive vesicles [52]. Depletion of Rab11a-positive vesicles in mouse oocytes leads to the release of these actin nucleators from vesicles, resulting in disorganization of cytosolic actin network [53]. We propose that the surface of long Osk and Yl-coated endocytic vesicles acts as platforms where the actin regulators promote the formation of long F-actin projections to anchor germ plasm components.

The yolk uptake-defective oocyte failed to maintain the localization of Kin-βgal, a microtubule plus-end marker, to the posterior cortex, resulting in the diffused localization of *osk* mRNA (Fig 3). In contrast, microtubule-dependent localization of Nod-βgal and *bcd* mRNA at the anterior or *grk* mRNA at the anterior-dorsal corner of the oocyte were all intact even in the absence of Yl-mediated endocytosis (Fig 3). These data indicate that effects caused by the malfunctioning of yolk uptake are limited in the posterior region. The posterior microtubule organization appears to be maintained by Osk-dependent recruitment of further microtubule plus ends [54]. Thus, the Yl-ligand axis may contribute to this Osk-dependent process.

Alternatively, the yolk uptake-dependent maintenance of microtubule organization at the oocyte posterior may be independent of Osk function. The posterior microtubule organization requires the plus-end-directed microtubule motor, Kinesin. It transports not only *osk* mRNA but also Dynactin to the posterior pole of the oocyte, which stabilizes the microtubule plus ends [55]. In addition, Kinesin drives bulk movement of the ooplasm during stages 10b to 12, known as ooplasmic streaming [56,57]. While the ooplasmic streaming is essential for the posterior accumulation of germ plasm components at late stages of oogenesis, its premature onset disrupts polarized microtubule arrays along the anterior-posterior axis, resulting in mislocalization of *osk* mRNA. The timing of the ooplasmic streaming is regulated by Capu and Spir [58]. Capu and Spir promote assembly of the ooplasmic actin mesh that is proposed to negatively regulate Kinesin [59,60]. Given the Capu- and Spir-mediated cortical actin remodeling, the ooplasmic actin mesh might also be disorganized in yolk uptake-defective oocytes, resulting in misregulation of the Kinesin-dependent processes to maintain posterior microtubule organization.

Intracellular localization of specific mRNAs in developing oocytes often occurs concurrently with vitellogenesis in oviparous vertebrates. In *Xenopus*, when the oocyte proceeds to the vitellogenic phase, several mRNAs including *Vg1* and *Dead end*, which are initially distributed throughout the cytoplasm in previtellogenic stage oocytes, start to be localized to the vegetal cortex [61,62]. Similarly, in zebrafish oocytes, several mRNAs show specific localization patterns, including localization to the animal pole (e.g., *zorba* and *c-mos)* [63,64] or to the vegetal pole (e.g., *mago nashi*) [65] during vitellogenic stages. The transport and anchoring of

*Xenopus Vg1* mRNA to the vegetal cortex are microtubule- and actin-dependent processes [66]. Interestingly, the *Xenopus Vg1* mRNA localization is disrupted by the depletion of vitellogenin [67]. Thus, the yolk uptake process appears to be linked to the cytoskeletal organization and mRNA localization in the vitellogenic-stage oocyte of diverse species.

## Materials and methods

### Flies

Flies were reared in vials containing a standard yeast-cornmeal-bran medium at 25°C. *y w* flies were used as wild-type controls. The transgenic stocks used were: *kin-lacZ* (*KZ503*) [3], *nod-lacZ* [38], *UASp-osk-bcd 3′UTR* [11], and *matα-GAL4-VP16* (Bloomington Stock Center).

### Cell line and culture

*Drosophila* S2 cells (a gift from Dr. H. Siomi, Keio University Medical School) were cultured in Schneider's *Drosophila* medium (Thermo Fisher Scientific, Waltham, Massachusetts, United States of America) containing 10% fetal bovine serum and antibiotics (100 U/ml penicillin and 100 μg/ml streptomycin) at 25°C in a humidified atmosphere.

### Plasmid constructs

*Pvas-yfp-vas* plasmids were generated as described [68]. For *pUASp-3×FLAG-GFP-osk-bcd 3′UTR* plasmids, coding sequences (CDSs) of long or short *osk* were PCR amplified using Q5 DNA polymerase (New England Biolabs, Ipswich, Massachusetts, USA) and were cloned into *pBluescript*. M139 of long *osk* was mutated to prevent translation of short Osk isoform which begins at M139 [5]. *3×FLAG-GFP* and a modified *bcd 3′UTR* [69] were joined to the 5′ and 3′ ends of *osk*, respectively. The resultant constructs were then subcloned into the *pUASp* vector. For *pCaSpeR-3×FLAG-GFP-osk* plasmids, a 6.5-kb *Xho*I-*Apa*I genomic DNA fragment encompassing the *osk* locus [5] was PCR amplified using Q5 DNA polymerase (New England Biolabs) and was cloned into *pBluescript*. To prevent translation of 1 of 2 Osk isoforms, an ATG triplet corresponding to the start codon for either isoform was mutated (M1L or M139L mutation to express only the short or long isoform, respectively) [5]. *3×FLAG-GFP* was then inserted at the 5′ end of each isoform. The resultant *3×FLAG-GFP-osk (M1L or M139L)* construct was then subcloned into the *Xho*I and *Not*I sites of the *pCaSpeR*. For *pAc-3×FLAG-GFP-long* or *short osk*, and *pAc-yl-mCherry (mChe)-V5* plasmids, the PCR-amplified CDSs of *long osk (M139L)*, *short osk*, *yl*, *3×FLAG-GFP*, or *mChe-V5* were digested by appropriate restriction enzymes, and the resultant fragments were subcloned into the *pAc5.1/V5-HisA* vector (Invitrogen, Carlsbad, California, USA). The integrity of constructs was verified by DNA sequencing.

### Transgenic fly generation

Transgenic flies were generated by P-element-mediated transformation [70] using *y w* flies as recipients.

### Mutant generation with CRISPR-Cas9 system

Mutants for *yl*, *yp1*, *yp2*, *yp3*, and *apolpp* were isolated using the CRISPR-Cas9 system as described [71]. Oligonucleotides for single guide RNA (sgRNA) were designed according to CRISPR Optimal Target Finder [72] (http://targetfinder.flycrispr.neuro.brown.edu/) to target 50 to 250 nt downstream of the start codon of the *yl*, *yp1–3*, or *apolpp* loci (see S2 Table for oligo pairs with sgRNA target sequences) and were ligated into the *pDCC6* vector [73] linearized with *Bbs*I. The plasmid was injected into *y w* eggs and the resulting adults were crossed

with balancer flies. CRISPR-Cas9-induced mutations in F1 males were detected through T7 endonuclease I assay [74] and direct sequencing of PCR products. *yp1* and *yp2* double mutants were generated by injecting with mixture of plasmids containing sgRNA sequence for *yp1* or *yp2*. Triple mutants for *yp1–3* were produced through crosses between *yp1 yp2* double mutants and *yp3* mutants.

## Hatching assay

Virgin females of each genetic background tested were mated with *y w* males for few days at 25°C. They were then transferred to a chamber with an apple juice-agar plate to lay eggs for 4 h. The plates with approximately 200 eggs were incubated at 25°C, then the number of hatched eggs was counted every 4 h from 16 to 20 h to 68 to 72 h after egg laying. The number of unhatched eggs was also counted at the last time point to calculate the hatching rate. Experiments were performed in 5 independent replicates. *y w* females were used as a control.

## Antibody generation

The CDS of the intracellular region of Yl (amino acid residues 1822–1984 of Yl) was cloned into the *pProExHTa* vector (Gibco BRL, Gaithersburg, Maryland, USA) to produce a 6×His-tagged protein. The fusion proteins were expressed in *E. coli* BL21 cells, were purified with Ni-NTA agarose (Qiagen, Hilden, Germany) under denaturing conditions, and were subjected to a disc preparative gel electrophoresis (Nihon Eido, Tokyo, Japan). The polyclonal antisera were raised in rabbits and rats by SCRUM Inc. (Tokyo, Japan). The polyclonal guinea pig antibodies against Rab5 were generated by SCRUM Inc. using the synthetic peptide, TSIRPTG-TETNRPTNNC (corresponding to amino acid residues 201–217 of Rab5) coupled with keyhole limpet hemocyanin (KLH). The polyclonal rabbit antibodies against long Osk were generated by MBL (Nagoya, Japan) using the synthetic peptide, MAAVTSEFPSKPISYTSC (corresponding to amino acid residues 1–17 of long Osk) coupled with KLH.

## Transfection

The *pAc-Yl-mChe-V5* plasmids were cotransfected with either *pAc-3×FLAG-GFP-long* or *short osk* plasmids into S2 cells using HilyMax reagent (Dojindo, Kumamoto, Japan). Transfection was conducted with 2.5 μg plasmid DNA and 5 μl transfection reagent per $2 \times 10^6$ cells. After cultured for 4 h, the medium was replaced by a fresh complete medium. Approximately 48 h posttransfection, cell culture was plated onto cover slips precoated with poly-L-lysine for 30 min and was fixed with 4% paraformaldehyde (PFA) in PBS at room temperature for 20 min. Fluorescent signals were observed under the confocal microscopy (Olympus FV1000-D).

## Immunofluorescence

Immunostaining of dissected ovaries was performed by standard procedures [11]. The primary antibodies used were: rabbit anti-Vas (lab stock, 1: 2,000), rat anti-Stau (lab stock, 1: 4,000), guinea pig anti-Osk (lab stock, 1: 1,000), rabbit anti-long Osk (1: 1,000), mouse anti-β-galactosidase (Promega, 1: 5,000), guinea pig anti-Rab5 (1: 5,000), affinity-purified rabbit anti-Rab11 [11] (1: 10,000), and rabbit anti-Tud [75] (1: 2,000). Alexa Flour 488- and 555-conjugated secondary antibodies (Thermo Fisher Scientific) were used at 1: 500. Fluorescent dye-conjugated phalloidin (Thermo Fisher Scientific or Abcam, Cambridge, United Kingdom) and DAPI were used to label F-actin and nuclei, respectively. After staining, samples were mounted in ProLong Diamond mounting media (Thermo Fisher Scientific). The FM4-64 incorporation assay was performed as described [11]. For immunostaining of embryos, embryos were

dechorionated with sodium hypochlorite and were fixed using 4:1 heptane and 4% PFA/PBS. The vitelline membrane was hand peeled with a tungsten needle without methanol treatment that disrupts F-actin. Vas and F-actin were stained with rabbit anti-Vas (lab stock, 1: 2,000) and Alexa Flour 647-conjugated phalloidin (Thermo Fisher Scientific), respectively. Confocal images were obtained using a laser confocal microscope (Olympus FV1000-D) with a 60× NA 1.2 UPLSAPO water immersion lens or a 100× NA 1.4 oil immersion lens (Olympus, Tokyo, Japan). Images were processed with Adobe Photoshop (Adobe Systems Incorporated, San Jose, California, USA).

## RNA in situ hybridization

Dissected ovaries were fixed in 4% PFA/PBS for 20 min at room temperature. After fixation and washes in PBST (0.2% Tween 20/PBS), 3 μg/ml proteinase K was applied for 12 min at room temperature. The digestion was stopped with 2 mg/ml glycine. The samples were washed twice with PBST and were refixed with 4% PFA/PBS for 20 min at room temperature. After washes in PBST, PBST was gradually replaced with hybridization buffer (50% formamide, 5×SSC (pH 5.0), 0.1 mg/ml salmon sperm DNA, 50 μg/ml heparin, 0.1% Tween 20), and the samples were prehybridized for 1 h at 55°C. Hybridization was carried out at 55°C overnight in hybridization buffer with RNA probes for *osk*, *bcd*, or *grk* synthesized in the presence of digoxigenin (DIG)-UTP (Roche, Penzberg, Germany). Excess probes were removed by 3 washes with 0.1% Tween 20/50% formamide/5×SSC for 10 min each at 55°C followed by 4 washes with 0.1% Tween 20/50% formamide/2×SSC for 30 min each at 55°C. Then, the wash buffer was gradually replaced with PBST. The hybridized DIG-labeled probes were detected with standard procedures using mouse anti-DIG (Roche, 1: 400) and Alexa-488-conjugated anti-mouse IgG (Thermo Fisher Scientific, 1: 500).

## Image analysis

For colocalization analyses of Osk isoforms with Yl, ovaries expressing 3×FLAG-GFP-long or short Osk were stained with anti-FLAG (Sigma, 1: 1,000) and rat anti-Yl (1: 5,000) followed by Alexa-488-conjugated anti-mouse IgG (1: 500) and Alexa-555-conjugated anti-rat IgG (1: 500), respectively. Samples were mounted in ProLong Glass mounting media (Thermo Fisher Scientific). Confocal images were obtained using a laser confocal microscope (Leica TCS SP8) with a 100× NA 1.46 oil immersion lens (Leica, Wetzlar, Germany) in a photon counting mode of the HyD detector and were deconvolved with Huygens Essential software (SVI). Pearson correlation coefficient for regions of interest (ROIs) in background-subtracted images was calculated using the Coloc 2 plugin in Fiji [76] (https://imagej.net/Fiji) with preset parameters. For quantification of YFP-Vas signals, z-section images (spaced by 1.5 μm over 20 to 35 μm) of posterior region of stage 14 oocytes were acquired using Leica TCS SP8. Signal intensity for ROIs was measured by Fiji software. A sum of intensity from multiple z-sections was assumed as the total YFP intensity per oocyte. For measuring signal intensity values for immunostained Stau and Osk, confocal images at the posterior region of stages 9 and 10b oocytes were obtained using Leica TCS SP8. Rectangular ROIs with 2 μm × 5 μm size were taken every 2 μm along the anterior-posterior axis from the posterior pole of the oocyte and mean intensities within each ROI were measured using Fiji software. Significant differences (*p*) between mutant and wild-type oocytes were calculated by Mann–Whitney *u* test.

## Immunoprecipitation and immunoblot analysis

Ovaries overexpressing 3×FLAG-GFP-long or short Osk at the anterior pole of the oocyte were lysed in lysis buffer (25 mM HEPES (pH 6.8), 150 mM KCl, 1 mM MgCl$_2$, 1% Triton X-

100, 250 mM sucrose, plus protease inhibitors) and were incubated on ice for 1 h. After centrifugation, the supernatant containing 5 mg of total protein was incubated with anti-FLAG M2-Sepharose (Sigma, St. Louis, Missouri, USA) overnight at 4˚C. These beads were then washed 5 times with lysis buffer. Bound proteins were eluted from the beads with lysis buffer containing 500 μg/mL 3×FLAG peptides. The eluted proteins were separated on 4% to 12% NuPAGE gels (Thermo Fisher Scientific) with MOPS buffer (Thermo Fisher Scientific), and were either silver stained using Pierce Silver Stain for Mass Spectrometry (Thermo Fisher Scientific) or transferred to Immobilon-P membrane (Millipore, Burlington, Massachusetts, USA). The membrane was subjected to western blot analysis with mouse anti-FLAG M2 (Sigma, 1: 10,000), mouse anti-α-Tubulin (Sigma, 1: 10,000), rabbit anti-Yl (1: 80,000), rat anti-Yl (1: 10,000), guinea pig anti-Rab5 (1: 50,000), affinity-purified rabbit anti-Rab7 [11] (1: 20,000), and affinity-purified rabbit anti-Rab11 [11] (1: 50,000), followed by HRP conjugated secondary antibodies (Jackson ImmunoResearch Laboratories Inc., West Grove, Pennsylvania, USA). The membrane was incubated with SuperSignal Dura (Thermo Fisher Scientific) and the chemiluminescent signals were detected with a LAS-3000 mini image analyzer (Fujifilm, Tokyo, Japan).

## Protein identification by mass spectrometry

The silver-stained protein bands were excised from the NuPAGE gels (Thermo Fisher Scientific), were cut into small pieces (approximately 1 mm$^3$), and were destained. Proteins of the gel pieces were reduced with DTT (Thermo Fisher Scientific), were alkylated with iodoacetamide (Thermo Fisher), and were digested with trypsin and lysyl endopeptidase (Promega, Madison, Wisconsin, USA). The resultant peptides were analyzed on an Advance UHPLC system (Michrom Bioresources, Auburn, California, USA) coupled to a Q Exactive mass spectrometer (Thermo Fisher Scientific) with the raw data processed using Xcalibur (Thermo Fisher Scientific). The raw data were analyzed against the SwissProt database or NCBI nonredundant protein database restricted to *Drosophila melanogaster* using Proteome Discoverer version 1.4 (Thermo Fisher Scientific) with the Mascot search engine version 2.4 (Matrix Science, London, UK). A decoy database comprised of either randomized or reversed sequences in the target database was used for false discovery rate (FDR) estimation, and Percolator algorithm was used to evaluate false positives. Search results were filtered against 1% global FDR for high confidence level.

## Supporting information

**S1 Fig. Association of long Osk with Yl.** (A, B) Signals for FLAG-GFP-tagged long or short Osk, Rab11, and Vas at the anterior region of the stage 10b oocyte expressing *osk-bcd 3′UTR*. DNA stained with DPAI is shown in cyan. Endosomal protein Rab11 and germ plasm component Vas were recruited by FLAG-GFP-tagged long and short Osk, respectively, at the anterior region of the oocyte. (C) A silver-stained gel of long or short Osk immunoprecipitates used for the mass spectrometry analysis. Lysates of control oocytes or oocytes expressing 3×FLAG-GFP-long or short Osk were immunoprecipitated using anti-FLAG antibodies. Boxes (labeled L1 to L8 for long Osk IP and S1 to S8 for short Osk IP) represent protein bands that were excised for the mass spectrometry analysis. (D, E) Localization of 3×FLAG-GFP-long or short Osk and Yl-mChe in S2 cells. S2 cells were cotransfected with plasmids that express 3×FLAG-GFP-long or short Osk and Yl-mChe. An uncropped gel image for panel C can be found in S1 Raw Images. Scale bars: 20 μm (A, B) or 5 μm (D, E). GFP, green fluorescent protein; IP, immunoprecipitation; mChe, mCherry; Osk, Oskar; Vas, Vasa; Yl, Yolkless. (TIF)

**S2 Fig. Generation of mutants for *yl* and its ligands.** (A) A schematic drawing of domain organization of Yl protein. Blue, gray, and green boxes indicate LDL receptor class A repeat, LDL receptor class B repeat, and EGF-like domain, respectively. (B) DNA sequences of *yl* gene of wild type and 2 *yl* alleles ($yl^{16-2}$ and $yl^{16-6}$). Target sites for sgRNA are shown in blue. The deletion and point mutation are shown in red. An annotated ATG, which does not match to canonical translation initiation consensus, and a downstream in-frame second ATG are shown in green. Numbering shown above DNA sequences is started with number 1 at the A of the initiation ATG codon. (C) Immunoblots for Yl of ovarian lysates from wild-type, hemizygous $yl^{16-2}/Df(1)KA9$, and $yl^{16-6}/Df(1)KA9$ females. The protein band of Yl at about 250 kDa was undetectable in *yl*-deficient mutants. α-Tub was used as a loading control. (D, E) DNA sequences of *yp1*, *yp2*, *yp3* (D), and *apolpp* (E) genes around the mutation sites. Target sites for sgRNA are shown in blue. Deleted bases are shown as red dashes. The ATG corresponding to translational initiation site is shown in green. Numbering shown above DNA sequences is started with number 1 at the A of the initiation ATG codon. Uncropped blot images for panel B can be found in S1 Raw Images. α-Tub, α-Tubulin; EGF, epidermal growth factor; LDL, low-density lipoprotein; sgRNA, single guide RNA; Yl, Yolkless.
(TIF)

**S3 Fig. YFP-Vas fails to accumulate to the posterior cortex in *yl* hemizygous oocytes.** Localization of YFP-Vas at the posterior region of oocytes in wild type (A, D), $yl^{16-2}/Df(1)KA9$ (B, E), and $yl^{16-6}/Df(1)KA9$ (C, F) at stages 10b and 14. Oocytes are outlined by blue dashed lines. Scale bars: 10 μm. Vas, Vasa; YFP, yellow fluorescent protein.
(TIF)

**S4 Fig. Yl and its ligands are required for posterior localization of *osk* mRNA.** Posterior region of stage 10b oocytes stained for *osk* mRNA and Stau in wild-type (A), $yl^{-/-}$ (B), and $yp1$–$3^{-/-}$ $apolpp^{-/+}$ (C) ovaries. Oocytes are outlined by blue dashed lines. Scale bars: 10 μm. Osk, Oskar; Stau, Staufen; Yl, Yolkless.
(TIF)

**S5 Fig. Yl and its ligands are dispensable for long Osk-mediated endocytic activation.** Rab5 (A-F) and Rab11 (G-I) at the anterior region of stage 10b oocytes without or with *osk-bcd 3'UTR* expression in wild-type (A, D, G), $yl^{-/-}$ (B, E, H), and $yp1$–$3^{-/-}$ $apolpp^{-/+}$ (C, F, I) ovaries. Scale bars: 20 μm. Osk, Oskar; Yl, Yolkless.
(TIF)

**S6 Fig. Roles of the yolk uptake in *osk* mRNA localization and cortical anchoring of germ plasm components.** At vitellogenic stages, Yl binds to yolk proteins (Yp1–3 and Apolpp) at the oocyte surface. Ligand-bound Yl is then internalized by endocytosis and is delivered to endosomes. The yolk uptake process is required for maintenance of microtubule alignment to localize *osk* mRNA and long Osk-mediated actin remodeling to anchor germ plasm components to the oocyte posterior cortex. See the Discussion for detail. Apolpp, Apolipophorin; Osk, Oskar; Yl, Yolkless.
(TIF)

**S1 Table. A list of immunoprecipitated proteins with long and short Osk identified by mass spectrometry analysis.**
(DOCX)

**S2 Table. Oligo DNA pairs for sgRNA expression from pDCC6 plasmid.**
(DOCX)

**S1 Data. All individual numerical data in the different figure panels.**
(XLSX)

**S1 Raw Images. Full gel and blot images.**
(PDF)

## Acknowledgments

We thank Sachiko Otsu for technical assistance, the Bloomington *Drosophila* Stock Center for fly strains, and Hiroko Sano, Rikako Iemura, and Paul Lasko for critical reading of the manuscript.

## Author Contributions

**Conceptualization:** Tsubasa Tanaka, Naoki Tani, Akira Nakamura.

**Data curation:** Tsubasa Tanaka, Naoki Tani, Akira Nakamura.

**Formal analysis:** Tsubasa Tanaka, Naoki Tani.

**Funding acquisition:** Tsubasa Tanaka, Akira Nakamura.

**Investigation:** Tsubasa Tanaka, Naoki Tani, Akira Nakamura.

**Methodology:** Tsubasa Tanaka, Naoki Tani, Akira Nakamura.

**Project administration:** Tsubasa Tanaka, Akira Nakamura.

**Resources:** Tsubasa Tanaka, Akira Nakamura.

**Supervision:** Akira Nakamura.

**Validation:** Tsubasa Tanaka, Naoki Tani, Akira Nakamura.

**Visualization:** Tsubasa Tanaka, Naoki Tani, Akira Nakamura.

**Writing – original draft:** Tsubasa Tanaka, Naoki Tani, Akira Nakamura.

**Writing – review & editing:** Tsubasa Tanaka, Naoki Tani, Akira Nakamura.

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
