## [Editor Report · Decision Letter 0]

5 Nov 2020

Dear Dr Tanaka, 

Thank you for submitting your manuscript entitled "Requirement of receptor-mediated yolk uptake in oskar mRNA localization and cortical anchorage of germ plasm components in the Drosophila oocyte" for consideration as a Research Article by PLOS Biology. Thank you also for your patience as we completed our editorial process, and please accept my apologies for the delay in providing you with our decision.

Your manuscript has now been evaluated by the PLOS Biology editorial staff as well as by an academic editor with relevant expertise and I am writing to let you know that we would like to send your submission out for external peer review. However, we would like to consider your manuscript as a Short Report and in that format we only allow 4 main figures, so you will have to make some of the current ones Supplementary figures when completing your submission. You will have to change the article type from the drop down menu. 

Before we can send your manuscript to reviewers, we also need you to complete your submission by providing the metadata that is required for full assessment. To this end, please login to Editorial Manager where you will find the paper in the 'Submissions Needing Revisions' folder on your homepage. Please click 'Revise Submission' from the Action Links and complete all additional questions in the submission questionnaire.

Please re-submit your manuscript within two working days, i.e. by Nov 09 2020 11:59PM.

Kind regards,

Ines

--

Ines Alvarez-Garcia, PhD

Senior Editor

PLOS Biology

---

## [Decision Letter · Decision Letter 1]

17 Dec 2020

Dear Dr Tanaka,

Thank you very much for submitting your manuscript "Requirement of receptor-mediated yolk uptake in oskar mRNA localization and cortical anchorage of germ plasm components in the Drosophila oocyte" for consideration as a Short Report at PLOS Biology. Your manuscript has been evaluated by the PLOS Biology editors, an Academic Editor with relevant expertise, and by two independent reviewers.

As you will see, the reviewers find the results novel and interesting, but they also raise several points that need to be addressed. Reviewer 1 would like you to extend the discussion to consider further implications of the mechanism proposed, and Reviewer 2 thinks the localisation phenotypes are quite subtle and need to be properly quantified. The reviewers also ask you to clarify several other points.

In light of the reviews (attached below), we are pleased to offer you the opportunity to address the comments from the reviewers in a revised version that we anticipate should not take you very long. Please also pay attention to our data requirements included below. We will then assess your revised manuscript and your response to the reviewers' comments and we may consult the reviewers again.

We expect to receive your revised manuscript within 1 month.

**IMPORTANT - SUBMITTING YOUR REVISION**

3. Resubmission Checklist

a) *Published Peer Review*

b) *PLOS Data Policy*

Please provide the data underlying the following figures, and make sure you mention in the corresponding figure legends where the data can be found:

Fig. 1E, O, P and Fig. 2D, G

d) *Blurb*

Please also provide a blurb which (if accepted) will be included in our weekly and monthly Electronic Table of Contents, sent out to readers of PLOS Biology, and may be used to promote your article in social media. The blurb should be about 30-40 words long and is subject to editorial changes. It should, without exaggeration, entice people to read your manuscript. It should not be redundant with the title and should not contain acronyms or abbreviations. For examples, view our author guidelines: https://journals.plos.org/plosbiology/s/revising-your-manuscript#loc-blurb

Sincerely,

Ines

--

Ines Alvarez-Garcia, PhD,

Senior Editor,

PLOS Biology

Reviewers’ comments

Rev. 1:

The manuscript by Tanaka et al. describes parts of the mechanism that links Drosophila posterior patterning to yolk uptake into the oocyte. One of the two Oskar (Osk) isoforms, short Osk, recruits the germplasm factors to the posterior region of the oocyte, and the other one, long Osk, attaches to the surface of endosomes at the posterior cortex and is needed for the tight posterior anchoring of the germplasm factors. At the same time, localized Osk also stimulates local endocytosis. Tanaka and colleagues now convincingly show that endocytosis of yolk proteins through the Yolkless (Yl) receptor is needed for this anchoring and for establishing the proper cytoskeletal structures in the posterior region. Yl usually imports the three yolk proteins Yp1-3. Lack of yl has the strongest effect on posterior yolk import, anchoring of posterior factors, and organization of the posterior cytoskeleton. Similarly, but apparently less efficiently, disrupting in the same animal all three Yp1-3 genes also affected the posterior anchoring of long Osk and the posterior patterning. Additionally, the authors found evidence that Apolipophorin polypeptides are imported with the yolk proteins, too, and they might contribute to this process similar to the yolk proteins Yp1-3. Their functional importance is, however, not entirely clear yet.

By using ectopic (anterior) localization of tagged Osk isoforms the authors can demonstrate nicely what long Osk is capable of doing. The isolation of the complex components looks impressive and it gave them a good lead for their further, functional studies, which they performed also with mutants in the yl gene and in the Yp1-3 genes. The yl mutant generally produced a stronger phenotype than the Yp1-3 triple mutant, which is consistent with there being at least one additional protein that uses the yl receptor and contributes to the patterning function of yl. This could be Apolipophorin. However, because the gene encoding it is essential, this was not tested in the same way as for the Yp1-3 genes. Nevertheless, the model regarding Apolipophorin presented by the authors is consistent with the presented results. What was missing from the presented results with the yl and the Yp1-3 mutants are tests that show whether the mutant phenotypes are indeed caused by the known mutation in the respective genes and not due to second site hits. Admittedly, from the data, there is little doubt about this, and introducing additional genetic elements (like rescue constructs) could be a difficult task. Have other attempts been tried like using deficiencies or RNAi constructs? Even in cases where two different lines were used (e.g. yl16-2/yl16-6), the genetic background of the two alleles is probably at least very similar.

Results/Discussion:

The results might suggest that endocytosis acts by producing higher numbers of Yl receptors on endosomes (either a higher concentration of Yl on endosomes or more endosomes) and that these receptors help anchoring sufficient posterior factors. Could this be added to the discussion? It would also be good to know to which Yl domain large Osk attaches. Considering the expected topology of Yl in endosome membranes, does this interaction site support the presented model?

With good arguments, the authors make their case for the potential importance of the endosomes as docking sites supporting microfilament formation (although Fig 4R' seems to show as much or even more, but diffuse, F-actin than 4P'). However, could there also be a connection to kinesin-dependent cytoplasmic streaming? What are the connections between the posterior microtubule organization defect and the actin defect and how is this possible interaction affected by the mutants described here? Could this be discussed as well?

A possibly interesting aspect of this work that might deserve some additional consideration and discussion is that this mechanism couples nutrient uptake with evolutionary fitness. Insufficient yolk uptake not only slows down development and reduces the hatching rate, but it also reduces the number of germline progenitors that are being formed at the posterior end (this is all shown in this manuscript). This mechanism might thus help to adjust fecundity to food availability.

Minor issues and typos:

In Figure 3A-F, it is not clear what the white posterior crescent signal reflects (probably Kin-B-Gal). Could a different LUT be used? With the exception of E, these panels are difficult to judge.

line 295: Fig 4A-B(-C): only the lateral and posterior regions display a reduction of FM4-64 signal. Not the anterior.

line 134: "This result is consistent…" is an isolated statement, and in the context of the RNA binding proteins just described to bind to Osk, it does not support a direct binding of Oskar to RNA.

Fig S2B: state what blue sequence means (see D, E)

line 238: "is probably caused" sounds too speculative and could be replaced by "might be caused"

line 381: completed (since "developed" on line 382)

line 411: rather than by/through a general decrease in (missing word)

Rev. 2:

In this manuscript Tanaka et al. identify the yolk receptor, Yolkless, as protein that is pulled-down with the long isoform of Oskar. They go onto show that yolkless mutants cause similar defects in the anchoring of the pole plasm and the organisation F-actin and microtubule cytoskeletons to the loss of this long Oskar isoform. They also identify apolipoprotein 1 as a novel yolk protein in Drosophila and show that removal of all yolk proteins and 50% of apolipoprotein 1 gives a similar phenotype to loss of yolkless. Thus, these results reveal an unexpected link between vitellogenesis and the formation of the pole plasm.

The data are well-presented and generally support the main conclusions, but the phenotype is rather subtle. Staufen and Oskar are less tightly localised to the posterior cortex and the yolk protein null; apol1 heterozygotes show a 30% reduction in the number of pole cells. Although the involvement of yolk endocytosis in polar plasm anchoring is an interesting twist, there are a number of issues that will need to be resolved before publication and I suspect that this work is more appropriate for PLoS Genetics than PLoS Biology.

Main points:

1) The localisation phenotypes need to be properly quantified. For example, the main text states that "We found that, in yl-/- oocytes, Stau did not thoroughly target to the posterior pole and was observed distant from the posterior cortex, resulting in diffuse localization of Osk", but Figures 2 and 3 do not show this. In Fig 3H, almost all Staufen forms a clear posterior crescent, with a slight haze in the cytoplasm. Although Oskar protein is slightly more diffuse in Figure 2I-M, it is still all very close to the posterior cortex and cannot accurately be described as "diffuse". Similarly "Stau also failed to target to the posterior pole of these oocytes" is a gross overstatement of the data shown in Figure 3 and H. What is needed is a plot of the distributions of Staufen and Oskar across the anterior-posterior axes of multiple oocytes.

2) Figure S5 is misleading as it implies that Yolk endocytosis is upstream of microtubule alignment and oskar mRNA localisation. The microtubules are normal initially and yl mutants only disrupt the maintenance of this organisation.

3) The localisation of kinesin-betaGal to the centre of the oocyte in Fig3E seems anomalous given that Staufen is pretty well localised, but it is hard to see kinesin-betaGal is also enriched at the posterior. This is another phenotype that would benefit from proper quantification.

---

## [Editor Report · Decision Letter 2]

19 Feb 2021

Dear Dr Tanaka,

Thank you for submitting your revised Short Report entitled "Requirement of receptor-mediated yolk uptake in oskar mRNA localization and cortical anchorage of germ plasm components in the Drosophila oocyte" for publication in PLOS Biology. I have now obtained advice from the Academic Editor and discussed the revision with the other editors. 

We will probably accept this manuscript for publication, provided you satisfactorily address the remaining requests (see below).

We would like you to consider a change in the title to make it clearer:

"Receptor-mediated yolk uptake is required for oskar mRNA localization and cortical anchorage of germ plasm components in the Drosophila oocyte"

We expect to receive your revised manuscript within one week. 

-  a cover letter that should detail your responses to any editorial requests.

*Published Peer Review History*

*Early Version*

Sincerely,

Ines

--

Ines Alvarez-Garcia, PhD,

Senior Editor,

PLOS Biology

BLURB

Please provide a blurb which will be included in our weekly and monthly Electronic Table of Contents, sent out to readers of PLOS Biology, and may be used to promote your article in social media. The blurb should be about 30-40 words long and is subject to editorial changes. It should, without exaggeration, entice people to read your manuscript. It should not be redundant with the title and should not contain acronyms or abbreviations. For examples, view our author guidelines: https://journals.plos.org/plosbiology/s/revising-your-manuscript#loc-blurb

---

## [Editor Report · Decision Letter 3]

11 Mar 2021

Dear Dr Tanaka,

On behalf of my colleagues and the Academic Editor, Simon Bullock, I am pleased to say that we can in principle offer to publish your Short Reports "Receptor-mediated yolk uptake is required for oskar mRNA localization and cortical anchorage of germ plasm components in the Drosophila oocyte" in PLOS Biology, provided you address any remaining formatting and reporting issues. These will be detailed in an email that will follow this letter and that you will usually receive within 2-3 business days, during which time no action is required from you. Please note that we will not be able to formally accept your manuscript and schedule it for publication until you have made the required changes.

PRESS

Thank you again for supporting Open Access publishing. We look forward to publishing your paper in PLOS Biology. 

Sincerely, 

Ines

--

Ines Alvarez-Garcia, PhD 

Senior Editor 

PLOS Biology